# Perinatal Tetrahydrocannabinol Compromises Maternal Care and Increases Litter Attrition in the Long–Evans Rat

**DOI:** 10.3390/toxics12050311

**Published:** 2024-04-26

**Authors:** Emma Carlson, Eric Teboul, Charlene Canale, Harper Coleman, Christina Angeliu, Karissa Garbarini, Vincent P. Markowski

**Affiliations:** 1Department of Psychology, State University of New York at Geneseo, One College Circle, Geneseo, NY 14454, USA; elc12@email.sc.edu (E.C.); charlenecanale13@gmail.com (C.C.); harper.coleman@nih.gov (H.C.); cgangeli@buffalo.edu (C.A.); kkgarbar@buffalo.edu (K.G.); 2Departments of Neurosurgery and Neuroscience, Brown University & Rhode Island Hospital, Providence, RI 02912, USA; eric_teboul@brown.edu

**Keywords:** tetrahydrocannabinol, THC, 11-OH-THC, perinatal exposure, behavioral toxicity, developmental toxicity, maternal behavior, pup retrieval, benchmark dose

## Abstract

The marijuana legalization trend in the U.S. will likely lead to increased use by younger adults during gestation and postpartum. The current study examined the hypothesis that delta-9-tetrahydrocannabinol (THC) would disrupt voluntary maternal care behaviors and negatively impact offspring development. Rat dams were gavaged with 0, 2, 5, or 10 mg/kg THC from the 1st day of gestation through the 21st postnatal day. Somatic growth and developmental milestones were measured in the offspring, and maternal pup retrieval tests were conducted on postnatal days 1, 3, and 5. THC did not affect body growth but produced transient delays in the righting reflex and eye opening in offspring. However, there was significant pup mortality due to impaired maternal care. Dams in all THC groups took significantly longer to retrieve their pups to the nest and often failed to retrieve any pups. Serum levels of THC and metabolites measured at this time were comparable to those in breastfeeding women who are chronic users. Benchmark doses associated with a 10% reduction of pup retrieval or increased pup mortality were 0.383 (BMDL 0.228) and 0.794 (BMDL 0.442) mg/kg THC, respectively. The current findings indicate that maternal care is an important and heretofore overlooked index of THC behavioral toxicity and should be included in future assessments of THC’s health risks.

## 1. Introduction

The general public is increasingly accepting of the potential medical benefits of some cannabinoids. Attitudes toward recreational cannabis are also softening. Many young adults of reproductive age report fewer negative attitudes towards cannabis, increased perceived positive effects, and less risk awareness [1]. Younger users often discount public health information that emphasizes harmful effects, viewing it as less credible [2]. Such changes suggest increased risk for future perinatal exposures as adolescents enter their reproductive period [3]. The risks to these age groups are likely to grow as cannabis becomes more widely available, its potency increases, and its cost decreases.

Findings from both clinical and animal studies have shown that females are uniquely sensitive to the disruptive effects of cannabinoids [4,5]. Pregnant women who use cannabis reportedly do so to manage morning sickness, modulate aches and pains, and improve sleep [6,7,8,9]. Heavy use throughout pregnancy, such as that associated with cannabis use disorder, is linked with preterm delivery, small size for gestational age, lower birth weight, moderately abnormal Apgar scores, and increased risk of infant death within one year [10,11,12,13].

As a recreational drug, cannabis is typically consumed in doses sufficiently high to produce acute behavioral effects in adults. It is well-known that exposure to developmental neurotoxicants during the perinatal period can perturb brain development and produce permanent behavioral impairments in exposed offspring at doses that might not affect adults. Clinical evidence suggests that prenatal cannabinoid exposure leads to tremors and exaggerated startle responses in infants and attention deficits and other executive function disruptions in older children [14,15,16,17]. Available animal studies indicate that perinatal cannabinoid exposure alters motor and emotional behavior, produces cognitive impairments, and enhances the sensitivity to other recreational drugs [18,19,20,21,22,23,24].

In addition to direct pharmacological threats to the developing fetus or breastfeeding infant, the THC (delta-9-tetrahydrocannabinol) contained in cannabis could also negatively impact development via compromised maternal care. Women who use cannabis during the postpartum period tend to be less educated, have lower incomes, be younger, and be more likely to believe cannabis is safe for breastfeeding and can reduce pain, discomfort, and depression [1,25]. A recent student in Oregon, where medical and recreational cannabis use is legal, found that breastfeeding mothers increased their cannabis use during the early postpartum period, with the majority reporting that their reasons were for pain management and/or anxiety and depression. The self-reported increase in weekly use was accompanied by an increase in breastmilk THC concentrations [26]. Since many aspects of maternal caregiving hinge on the mother’s perception and processing of infant cues, the sensory distortions and psychomotor impairments produced by THC could disrupt infant care.

The belief that cannabis use during pregnancy and/or the postpartum period can ameliorate symptoms of depression is particularly alarming. While cannabis provides immediate relief from some depressive symptoms, its effects are transient and long-term use actually worsens the course of depressive disorders [27,28]. Women who use cannabis during the postpartum period are more likely to report depressive symptoms and breastfeed for shorter periods of time [29]. 

Despite the emerging evidence that cannabis can reduce breastfeeding and increase anxiety and depression in women, very few animal models have examined the effects of cannabinoids such as THC on maternal behavior. Well-designed animal models offer the opportunity to control many of the variables (age at exposure, dose, duration, medical history, etc.) that blur cause-effect relationships in epidemiological studies. Although maternal care behaviors in mammals are species-specific, there are shared underlying neuroendocrine mechanisms. For instance, in both rodents and humans, hormonal events over the course of pregnancy and the early postpartum period prime the mesolimbic dopamine (DA) system to respond to newborn cues, leading to the emergence of maternal behavior. In the female rat, an intact hypothalamic medial preoptic area (MPOA) is required for the onset and maintenance of maternal care behaviors such as nest construction, retrieving pups to the nest, and nursing them [30]. Maternal care is initiated when estrogen receptors in the MPOA are activated by estradiol during the late gestational period, ensuring that the dam is immediately responsive to her altricial pups. Later, maternal care is maintained by the sensory stimulation generated by newborn pups [31]. In the postpartum female rat, pup retrieval is a voluntary, proactive response mediated by the forebrain, while nursing is more reflexive and regulated by brainstem mechanisms. Activation of the mesolimbic DA system is known to promote goal-directed behavior and neurons in the MPOA project to the ventral tegmental area (VTA) and regulate the motivation to approach and nurture offspring [32,33]. 

Although rodent nest construction and pup retrieval are not directly translational, there are appetitive maternal behaviors in women that are also mediated by the mesolimbic DA system. Functional MRI studies in women have found that viewing pictures of their infant or listening to the sound of their own infant’s cry activates components of the mesolimbic DA system. A first-time pregnancy has been shown to produce volumetric changes in the ventral striatum, a key structure in the DA reward pathway. The structural changes are predictive of increased postpartum functional responses to infant cues such as pictures of women’s babies [34]. The ventral striatal response in mothers to pictures of their own children’s faces are also positively correlated with their plasma levels of oxytocin [35]. Although plasma levels of oxytocin are poor indicators of central activity, research with parturient rats has shown that MPOA neurons express oxytocin receptors following gestational priming with estradiol, and systemic oxytocin antagonists have been shown to impair pup retrieval [36]. The VTA also contains oxytocin receptors, and oxytocin has been thought to facilitate DA release in the nucleus accumbens in response to pup stimuli [37]. Although little research has examined the THC–oxytocin interaction, subchronic THC has been shown to selectively decrease oxytocin immunoreactivity in the rat VTA and nucleus accumbens [38].

The current study was designed to examine the long-term effects of perinatal exposure to THC on the social, cognitive, and motor behaviors of rat offspring throughout their lifespan. What the authors report here are transient developmental effects noted during the lactational period that emerged alongside dose-related maternal behavioral toxicity, an outcome that resulted in significant pup mortality. The maternal dose range of 0, 2, 5, or 10 mg/kg THC that was used in the current study incorporated NIDA’s “Standard THC Unit” of 5 mg/kg [39] and was based on earlier studies that found long-term behavioral deficits in offspring with little indication of maternal toxicity. For instance, 2 or 5 mg/kg was previously shown to increase locomotion in adult female and adolescent male offspring [19,40] and alter sociosexual approach behavior [20,41] and corticosterone activity in both sexes [22]. The 5 mg/kg dose also increased morphine self-administration and mu opioid receptor density in the VTA of female rat offspring [24], reduced the analgesic response to morphine in adult male offspring [23], and produced reference memory deficits in cognitive tasks [40,41]. Higher doses of 20 mg/kg also had sex-specific effects on sociosexual approach behavior, mesolimbic DOPAC levels, and striatal D1 and D2 receptor density in offspring [20,42]. Although none of these earlier studies reported altered maternal care behaviors, it is unclear if they were designed to examine such outcomes.

Maternal behavioral toxicity is typically regarded as a confounding factor in developmental neurotoxicity studies, and procedures such as pup cross fostering are used to eliminate its impact [43]. However, THC and other cannabinoids are not environmental toxicants but recreational drugs that are consumed voluntarily and in doses sufficiently high to alter adult behavior. The regulation of the mesolimbic DA system by the hormonally primed MPOA, as well as this circuit’s modulation by central oxytocin, could be disrupted by exogenous THC. Since maternal care is a robust, reward-mediated behavior in healthy dams [44], the hypothesis of the current study was that the acute effects of THC would supplant this natural reinforcer, resulting in less time and care directed toward offspring.

In the current study, pup retrieval tests were conducted during the early postpartum period following maternal THC administration. Because the dose range did not identify a no-observed adverse effect level (NOAEL) for THC, USEPA Benchmark Dose software(https://bmdsonline.epa.gov/, accessed on 20 April 2024) was used to determine a benchmark dose (BMD). The BMD is a range of chemical doses or concentrations that produce a predetermined change in the response rate of an adverse effect, in this case, a 10% increased risk of maternal behavioral toxicity. Blood samples from a group of dams were collected 60 min after THC administration to determine THC and metabolite concentrations at a time point when significant pup retrieval impairments were observed. 

## 2. Materials and Methods

### 2.1. THC Dosing Solution Preparation

The NIDA Drug Supply Program (https://nida.nih.gov/research/research-data-measures-resources/nida-drug-supply-program, accessed on 20 April 2024) provided the THC ((−)-trans-Delta-9-tetrahydrocannabinol, catalog #7370-001) for this research. The THC arrived dissolved in ethyl alcohol (500 g/5 mL) and packaged in ampules, the contents of which were mixed with 3 different volumes of sesame oil (Acros Organics, Thermo Scientific Chemicals, Waltham, MA, USA) to produce stock dosing solutions with concentrations of 12.5 mg/mL, 6.25 mg/mL, and 2.5 mg/mL. The stock solutions were then run on a rotary evaporator instrument in the SUNY Geneseo Biochemistry department (Geneseo, NY, USA) to remove the ethanol. 

### 2.2. Breeding and THC Administration

Adult male and female Long–Evans rats were purchased from Charles River (Wilmington, MA, USA). Rats were allowed to acclimate to the vivarium quarters for at least one week before breeding. Females were then housed with stud males and examined each morning (0830) for the presence of sperm upon vaginal smear. The sperm-positive day was regarded as gestation day (GD) 0. Sperm-positive rats were placed individually into polycarbonate cages (27 cm W × 48 cm L × 20 cm H) and assigned to a THC dose condition according to a randomized block design. Maternal body weights were recorded daily during the gestational period (see Figure 1). Rats were fed a nutritionally complete high-protein liquid diet formulated for pregnant/weanling rats (Lieber-DeCarli ′82, BioServ, Flemington, NJ, USA) throughout pregnancy and the lactation period. The liquid diet was mixed fresh every morning and provided in calibrated liquid feeding tubes to track daily intake. Water was available ad libitum. Illumination in the animal housing rooms was provided in a reverse 12 h light/12 h dark cycle in a barrier facility with an ambient temperature of 20 ± 2 °C and 40–60% humidity. All animal procedures complied with NIH guidelines [45] and were reviewed and approved by the SUNY Geneseo Institutional Animal Care and Use Committee (IACUC Protocols #2017-18-01 and 2020-21-01). Animal care and welfare were supervised by a licensed veterinarian.

Beginning on GD1, pregnant rats were gavaged once per day with 0, 2, 5 or 10 mg THC/kg BW in sesame oil. The concentrations of the dosing solutions were such that the daily dosing volume did not exceed 1.5 mL. Daily dosing continued until postnatal day (PND) 21 but not on the day of parturition. Because THC can be an anorectic in rats, a pair-fed control group was also created. Each dam in this group was yoked to one of the females in the 10 mg/kg group, and she was provided with the same volume of liquid diet consumed by her 10 mg/kg partner on the same gestational day. After weaning, offspring were fed standard pellet chow (Purina 5001 Rodent Diet, Nestlé Holdings, Inc., St. Louis, MO, USA). 

### 2.3. Developmental Milestones

Cages were inspected each morning for the presence of litters, and the day of parturition was regarded as PND0. Litters were culled to eight pups on PND5 using a randomized selection procedure. Four male and four female pups were kept from each litter whenever possible. Litter size, sex distribution, pup body weights, crown–rump lengths, and anogenital distances were recorded every other day from PND1 to 21. The righting reflex was measured in all pups on PND1, 3, 5, and 7. The righting reflex was defined as the latency (sec) in re-establishing posture (ventral surface down, all 4 paws in contact with cage floor) after being rolled on the left and right sides. The age of eye opening was examined on PND 13, 15, and 17. After weaning on PND21, offspring were housed in pairs with same-sex littermates.

### 2.4. Maternal Pup Retrieval

On PND1, 3, and 5, dams were dosed with THC as usual. One hour later, their pups were moved to the point furthest from the nest in the homecage, and their retrieval behavior was recorded with an overhead camera (Reolink RLC-810A, Wan Chai, Hong Kong Island, Hong Kong, China) at 25 frames per second. Retrieval tests lasted for 30 min or until the dam retrieved all of her pups back to the nest. All testing was conducted under dim red-light illumination during the dark phase of the animals’ cycle.

Videos were later scored by blind observers, and Python code utilizing DeepLabCut 2D markerless pose estimation was used to quantify locomotor activity during the 30 min tests. The pose estimation model was trained on a subset of recordings capturing the video capture variability across recordings before assessment of the complete dataset. Averaged positions of the nape and nose on the dams were used to estimate the leading portion (top half) of the rat, indicating directional positioning. X and Y coordinates were generated for the body parts being tracked in each video across time. The distance traveled by dams was converted from pixels to millimeters by applying video-specific scaling factors to cage dimensions. Averages of the X and Y coordinate data were used to quantify distance traveled. The data were then plotted to illustrate dam positioning.

### 2.5. Blood Sampling and THC Assays

On PND5, immediately after the pup retrieval test, dams with undersized litters (<8 pups) or litters with an uneven sex distribution of pups were euthanized with an overdose of sodium pentobarbital, trunk blood was collected and centrifuged, and the supernatant was frozen at −80 °C. Samples were shipped frozen overnight to the Texas A&M Veterinary Medical Diagnostic Laboratory (TVMDL) where they were assayed for THC, 11-hydroxy-THC, and 11-carboxy-THC levels via LC-MS/MS with protein precipitation. The detection limit was 10 ppb with samples ≥ 1 mL.

### 2.6. Statistical Methods

All repeated measurements were analyzed with repeated-measures analysis of variance (ANOVA) with PROC GLM SAS version 9.4 (SAS Institute Inc., Cary, NC, USA). The litter or rat dam served as the statistical unit of analysis. The THC dose condition and in some cases pup sex were between-subjects factors, and the PND was a within-subject factor. The Huynh–Feldt adjustment was used when appropriate. Newman–Keuls multiple-range tests were used to make pairwise comparisons. A *p* ≤ 0.05 was considered statistically significant.

## 3. Results

### 3.1. Gestational Body Weight

A repeated-measures ANOVA was conducted on the daily maternal body weight data from gestation day (GD) 1–20. There were significant main effects of THC dose [F(4,45) = 5.31, *p* = 0.0014] and GD [F(19,855) = 231.42, *p* < 0.0001] on body weight. Although the THC-by-GD interaction was not significant (*p* = 0.2), Newman–Keuls pairwise comparisons indicated that THC dose group differences emerged early in gestation, with the 10 mg/kg dams weighing less than controls from GD2–20. The 10 mg/kg dams also weighed less than each of the other dose groups on GD5, 7, 13, 17, 19 and 20. There were no differences between any of the other dose groups, including the control vs. the pair-fed group.

### 3.2. Serum Levels of THC and Metabolites in Dams

Because the sample sizes are small, individual data for serum levels of THC, 11-COOH-THC, and 11-OH-THC are presented in Figure 2. No blood samples were collected from control or pair-fed dams. Visual inspection of the data indicated that THC levels in the 2 mg/kg group, measured 1 h after dosing, were very low (although metabolites were reliably detected). 

### 3.3. Maternal Pup Retrieval

Pup retrieval latency (sec) data on PND1, 3, and 5 was analyzed with a repeated-measures ANOVA. A latency value of 1800 s was inserted for females that did not retrieve any pups during the 30 min tests. There was a significant main effect of THC dose [F(4,26) = 6.75, *p* = 0.0007; see Figure 3]. Since there were no effects of PND, the PND1, 3, and 5 data were averaged prior to conducting the pairwise comparisons. The Newman–Keuls test indicated that the control group latency was less than the 2, 5, and 10 mg/kg THC groups and that the pair-fed and 2 mg/kg groups were less than the 5 and 10 mg/kg groups.

Since the retrieval latency data were not normally distributed, they were also analyzed with the Kruskal–Wallis non-parametric one-way procedure (SAS 9.4, NPAR1WAY). Three analyses were conducted on PND1 and PND3, and the PND1, 3, and 5 data were averaged. There was a significant effect of THC dose (*p* < 0.02) in all three cases. 

The activity tracker was used to determine the distance traveled by the dams during the pup retrieval tests, and these data were analyzed in two different ways with a repeated-measures ANOVA. For the first analysis, the data were converted to an activity rate (m/min) because the retrieval test sessions were of different durations, ending either at 30 min or as soon as each dam retrieved all of her pups. There were no significant effects of THC on the activity rate (see Figure 4). Likewise, the Kruskal–Wallis found no significant effects of THC on the activity rates. 

For the second analysis, the distance traveled per number of pups retrieved was examined. This variable was examined because the litters were not culled until PND5 and there were different numbers of pups in each cage. Theoretically, retrieving more pups would require more travel distance. Again, there were no significant effects of THC at any of the PNDs according to a repeated-measures ANOVA as well as the Kruskal–Wallis test.

### 3.4. Effects of THC on Pup Developmental Milestones

#### 3.4.1. Body Growth

There were no significant effects of THC on the number of live pups born on PND0 or the sex distribution of the pups. Pups were weighed daily from PND1–21 and the body weights (g) for male and female pups were analyzed separately with a repeated-measures ANOVA. As expected, there were significant main effects of PND, as the pups grew throughout the lactational period, but there were no effects of THC. Crown–rump lengths (mm) were also measured throughout this period, and again, there were significant main effects of PND, but no effects of THC. Finally, the anogenital distance (mm) was measured on PND7, 15, and 21 and, as with body weight and crown–rump length, there were significant main effects of PND in both males and females but no effect of THC. 

#### 3.4.2. Eye Opening

Each pup was scored 0, 1, or 2 (no eyes open/unsealed, one eye, or both eyes, respectively), and a male and female litter mean was calculated at each PND. For the males, there were significant main effects of THC [F(4,19) = 4.83, *p* = 0.007], PND [F(2,38) = 132.55, *p* < 0.0001], and a THC-by-PND interaction [F(8,38) = 3.33, *p* = 0.007; see Figure 5]. Pairwise comparisons indicated that eye opening was transiently delayed on PND15 in the 10 mg/kg males compared to all other groups. For the females, there were significant main effects of THC [F(4,22) = 3.72, *p* = 0.02] and PND [F(2,44) = 150.96, *p* < 0.0001]. Again, there was a transient developmental delay, where the pair-fed, 5 mg/kg, and 10 mg/kg groups lagged behind the control and the 2 mg/kg groups on PND15.

#### 3.4.3. Righting Reflex

Right and left side latencies were averaged prior to analysis with a repeated-measures ANOVA. THC dose served as a between-subjects factor and sex and PND as within-subject factors. There were no effects of sex but there was a significant THC-by-PND interaction [F(12,147) = 2.16, *p* = 0.03; see Figure 6], where the 10 mg/kg group’s latency was significantly longer than all other groups on PND1.

### 3.5. Effects of Impaired Maternal Care on Pup Mortality

Litter survival was significantly compromised by maternal THC administration during the PND0–8 period (see Figure 7). To examine this outcome, data were treated as binary, where code 1 represented litters where all pups survived and code 0 represented entire litters that were lost due to maternal neglect or infanticide. There was a significant effect of THC on survival [F(4,48) = 4.66, *p* = 0.003], where litters in the 2, 5, and 10 mg/kg groups had a lower survival mean than control and pair-fed litters. Interestingly, there were no further instances of infanticide in THC-dosed litters after PND8.

### 3.6. Benchmark Dose Analysis

The infanticide data were unanticipated and consequential for this study. Since the dose–response data did not identify a NOAEL (no-observed adverse effect level), USEPA Benchmark Dose Software (BMDS Online, Build 2BE68256DAEE; Model Library Version 2021.09) was used to provide additional information about the dose–response relationships in the pup retrieval and infanticide data. Because both variables were binary, where dams either retrieved all of their pups or none of them, or where the dams killed all of their pups or killed none of them, BMD dichotomous modeling was used. For pup retrieval, the Multistage 1° model was recommended as the best-fitting model (see Figure 8) with the lowest Akaiki Information Criterion (AIC). The BMD and BMDL associated with a 10% risk of pup retrieval failure generated by this model were 0.383 and 0.228 mg/kg THC, respectively. For infanticide, the LogLogistic model was the recommended best fitting model (see Figure 9). The BMD and BMDL associated with a 10% risk of infanticide generated by this model were 0.794 and 0.442 mg/kg THC, respectively.

## 4. Discussion

The current study found that a single daily oral dose of 5 or 10 mg/kg THC, administered to rat dams from GD1-PND21, produced transient delays in offspring eye opening and the righting reflex but did not affect any indices of body growth. However, our assessment of developmental toxicity is likely conservative since a significant number of litters in these dose groups were lost as a consequence of impaired maternal care. Dams that received 2, 5, or 10 mg/kg THC were significantly less likely to retrieve their pups to the nest when tested 1 h after THC administration on PND1, 3, and 5. The pup retrieval failures were representative of other maternal care deficits observed in the home cage. Instead of nesting and nursing, dams ignored their pups and engaged in purposeless motor behavior. Pup death did not result from maternal aggression or cannibalism. Pups were often found scattered throughout the cage or buried in piles of bedding. The first 8 postnatal days appeared to be a period of increased risk since no further pup deaths occurred after this time.

Very few animal studies have examined the effects of THC on maternal behavior despite the fact that it is a psychoactive substance that is consumed in doses sufficient to induce euphoria, sensory distortions, and psychomotor impairments. However, maternal care deficits have been noted in some older animal studies. Abel [46] found that doses of 10 or 20 mg/kg THC administered to rats on PND7 or 10 reduced pup retrieval in a dose-related fashion. Although Abel’s dose of 20 mg/kg is higher than what was used in the current study, it was an acute treatment, whereas the dams in the current study likely experienced bioaccumulation. Other studies from this period reported significant postnatal pup mortality but often attributed it to THC-induced maternal nutrition deficits [47,48]. 

The current findings argue against cannibalism triggered by THC-induced nutritional deficits. A pair-fed control group was yoked to the 10 mg/kg THC group and there were no pup retrieval deficits or postnatal deaths in this group. Although the dams in the 10 mg/kg group weighed less throughout gestation, their pups did not. Others have shown that rats that experience caloric restriction during gestation and lactation actually build better nests and have shorter latencies during pup retrieval tests [49]. Furthermore, there were no maternal body weight differences in the 2 and 5 mg/kg groups despite their significant postnatal pup loss and retrieval deficits. Instead, the behavior of the THC dams resembled what Numan et al. [33] described during pup retrieval tests following micro-injection of the GABA_a_ agonist drug, muscimol, into the VTA. In both cases, females ignored their pups and engaged in aimless motor stereotypy, sniffing, and rearing. These authors interpreted the muscimol effects as a consequence of dopaminergic hyperexcitation via disinhibition. Micro-injection of a GABA_b_ agonist, baclofen, disrupted pup retrieval but also produced general locomotor impairment. However, the activity tracker data in the current study did not detect differences in locomotor output across the dose groups. Mice with inactivated GABA_a_ receptors due to reduced expression of the delta subunit also displayed disrupted pup retrieval and nest construction. These effects were severe enough to increase the pup mortality rate during the first 24 h of the postpartum period due to maternal neglect or cannibalism [50]. Other manipulations that disrupt the MPOA to mesolimbic circuit that controls appetitive maternal behavior impair pup retrieval as well [30]. For instance, microinjection of a D1 receptor antagonist into the NAcc impaired pup retrieval at doses that did not affect general locomotor activity [51]. 

The current authors are not arguing that pup mortality was directly due to nest retrieval failures but rather a more general disruption of voluntary, appetitive maternal care behaviors mediated by the MPOA–mesolimbic DA circuit. It is reasonable to assume that maternal care was compromised for a period of several hours post-dosing once THC brain concentrations exceeded an effect threshold. For instance, microinjection of a D1 antagonist into the VTA produced a transient pup retrieval deficit that was no longer present after 5 h. These pup retrieval effects were not due to impaired locomotion [51]. Others have shown that THC blood levels rise to a peak within 30 min and are sustained for 4 h following intraperitoneal administration to female rats [52]. It is also possible in a chronic exposure study like ours that THC withdrawal could compromise maternal care. However, these assumptions require a more detailed analysis of pup retrieval at different time points following dosing.

The mechanisms responsible for the delayed development of the righting reflex and eye opening are unknown, but they could be a direct effect of THC on the offspring rather than a secondary consequence of impaired maternal care. Although periods of maternal separation have wide-ranging impacts on open-field, anxiety, and somatosensory behaviors in rat offspring, they do not affect the righting reflex [53]. Dams that consume a low-protein diet throughout gestation deliver pups that show righting reflex and eye opening delays, but these effects are accompanied by body weight and somatic growth deficits [54] that we did not observe in the THC pups. Furthermore, we fed our dams with a high-protein diet that was designed for pregnant and weanling rats. Given that the righting reflex delay was only observed in neonates (PND1), the THC effect was likely localized to the spinal cord or medulla since ontogenic studies have shown that the righting reflex becomes encephalized with advancing age [55]. Di Bartolomeo [56] observed similar righting reflex delays on PND1 following maternal doses of 5 mg/kg from GD15-PND9. The reflex delays could be a consequence of altered endocannabinoid and dopamine signaling since brain 2-AG levels were decreased, the metabolic enzyme monoacylglycerol lipase was elevated, and D2 receptor expression was increased in these offspring.

Our blood sampling procedure found that serum THC concentrations in the 5 and 10 mg/kg groups were comparable at 1 h following dosing, which was the same time the pup retrieval tests were conducted. Although the sample size in the 10 mg/kg group is small and the results should be viewed as tentative, the metabolite levels in this group were much higher, in line with an earlier study that found greater uptake of 11-OH-THC vs. THC into the male rat brain [57]. The serum THC concentration in the 2 mg/kg group was very low despite the significant pup retrieval impairment, suggesting that 11-OH-THC might be functioning as an active metabolite or that brain THC concentrations were higher than serum levels. Hutchings et al. [58] administered oral doses of 15 or 50 mg/kg THC to pregnant rats and found that consecutive daily dosing throughout the last two weeks of gestation led to significant accumulation in the dam and fetus compared to a single acute dose. Since THC can be sequestered in fatty tissues such as the brain or breastmilk [59], our serum measurements might be lower than brain levels. 

The THC concentrations in the 5 and 10 mg/kg rat dams are similar to the plasma levels measured in a group of breastfeeding women (median of 3.7 ng/mL) who were chronic daily cannabis users. Breastmilk levels in these women were 7× higher than plasma levels, suggesting bioaccumulation and enhanced lactational transfer [26]. It would be useful to measure breastmilk levels in an animal study like ours to determine if the milk/plasma ratio is comparable to that reported in women. Interestingly, two of the participants in the study of Moss et al. [26] consumed edibles prior to sample collection, and their samples were in the same range as the rest of the study sample who were consuming via inhalation. Nadulski et al. [60] conducted a more direct examination of oral kinetics in humans following consumption of a THC gelcap. They reported a mean peak plasma level of 3.19 ng/mL at 64 min post administration. Again, these human data are very similar to the current rat data. 

The current study has several limitations that should be addressed in future investigations. Importantly, we did not examine other rat maternal behaviors such as pup licking and grooming, nursing posture, or the amount of time spent nursing. Additionally, although we anecdotally observed THC-dosed dams burying their pups in piles of bedding, engaging in repetitive self-grooming, and sleeping away from their litters, we were not able quantify these behaviors in a consistent manner. Future studies should include a fine-grained analysis of what the THC dams were doing in their home cage instead of caring for their pups. Furthermore, we were not able to determine if the effects on pup righting reflex and eye opening were a direct effect of THC or a consequence of impaired maternal care. If the pup effects were due to a direct effect of prenatal THC exposure, a cross-fostering procedure would be informative. Finally, more effort should be directed toward determining the neuroendocrine mechanisms of the maternal behavioral toxicity. Useful first steps might include serum corticosterone and oxytocin levels as well as microdialysis assessment of hypothalamic and mesolimbic dopamine activity at different time points after THC ingestion.

We were unable to find any published BMD analyses for THC using pup mortality or maternal rat data such as ours. However, DeSesso et al. [61] reviewed an earlier BMD analysis of dimethoate, an organophosphate pesticide, which was based in part on neonatal rat pup mortality. In the dimethoate study, pup deaths initially appeared to be the most sensitive endpoint until it was revealed that the deaths were clustered in a small number of litters and resulted from maternal care deficits. Like the current study, pup death occurred during the first few postnatal days despite the lack of apparent developmental toxicity in neonates at birth. When the dimethoate data were later reevaluated using the litter as the statistical unit of analysis rather than individual pups, maternal behavioral toxicity emerged as a more appropriate endpoint for risk assessment. In the current THC study, the litter was used as the statistical unit of analysis, and the pup deaths were dichotomous, affecting every member of the litter or none. 

Hindelang et al. [62] used an acute increase in blood pressure in mice as an endpoint and derived a BMD of 3.05 mg/kg THC. However, these authors commented on the dearth of appropriate dose–response data from animal studies that featured chronic THC administration and examined relevant toxicological endpoints. Lachenmeier and Rehm [63] used a margin of exposure (MOE) risk assessment approach and concluded that cannabis is in the “low-risk range” for humans. However, the MOE assessment was based on a single, very high lethal dose that was acutely administered to adult rats. The data presented in the current study suggests that the MOE approach does not adequately capture the risk presented to the mother–infant dyad by chronic lower-dose THC exposures. The authors further suggest that the NIDA standard THC unit of 5 mg/kg should anchor the high end of the dose–response relationship in future studies of the mother–infant dyad. 

## Figures and Tables

**Figure 1 toxics-12-00311-f001:**
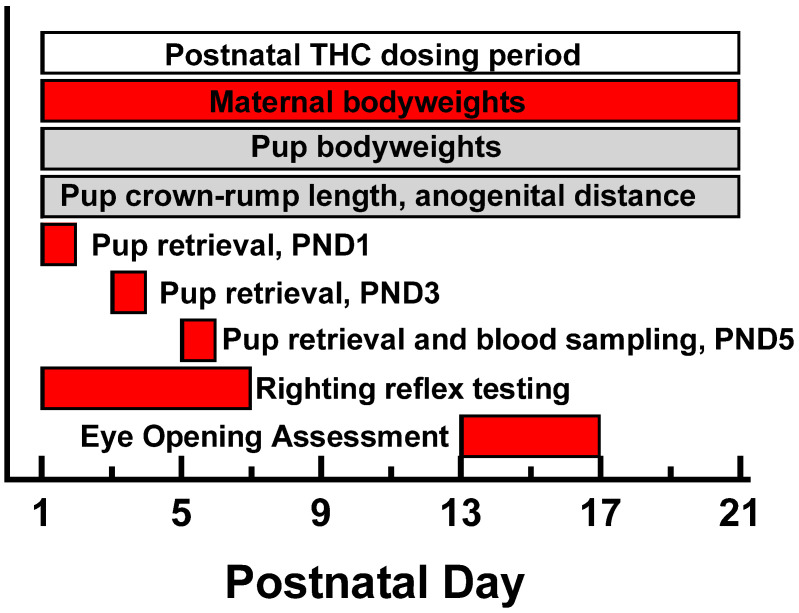
Timeline of major events during the postnatal maternal THC dosing and data collection period. Bars in red indicate events that were associated with significant THC effects. Bars in gray were not significant.

**Figure 2 toxics-12-00311-f002:**
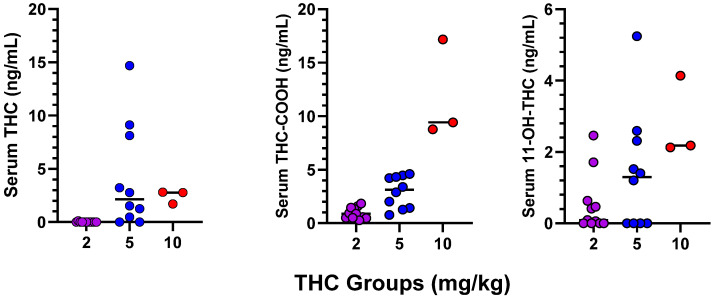
Serum levels of THC and metabolites in rat dams, collected on postnatal day 5 immediately after the pup retrieval test.

**Figure 3 toxics-12-00311-f003:**
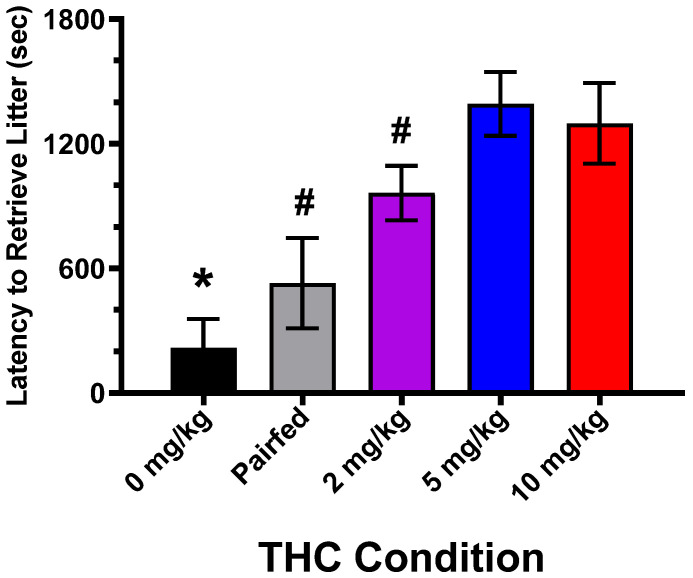
Mean latency (±SEM) in retrieving all pups to the home cage nest site during the 30 min tests. Latency data from postnatal days 1, 3, and 5 were averaged. * 0 mg/kg < 2, 5, and 10 mg/kg, # Pair-fed and 2 mg/kg < 5 and 10 mg/kg. N = 8, 6, 10, 8, 6, for the 0 mg/kg, pair-fed, 2 mg/kg, 5 mg/kg, and 10 mg/kg groups, respectively.

**Figure 4 toxics-12-00311-f004:**
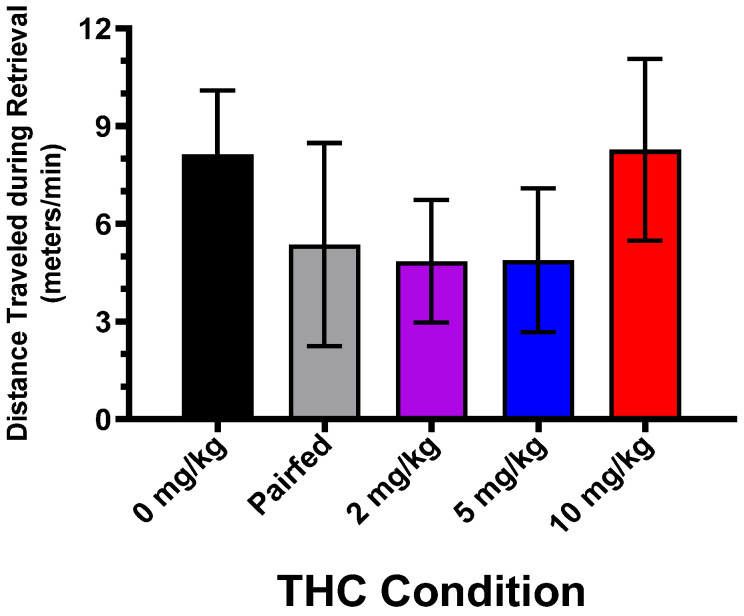
Mean activity rate (±SEM) for the rat dams during the 30 min pup retrieval tests. Activity data from postnatal days 1, 3, and 5 were averaged. There were no differences between the groups, indicating that the effects on pup retrieval were not due to altered locomotor behavior. N = 8, 6, 10, 8, 6, for the 0 mg/kg, pair-fed, 2 mg/kg, 5 mg/kg, and 10 mg/kg groups, respectively.

**Figure 5 toxics-12-00311-f005:**
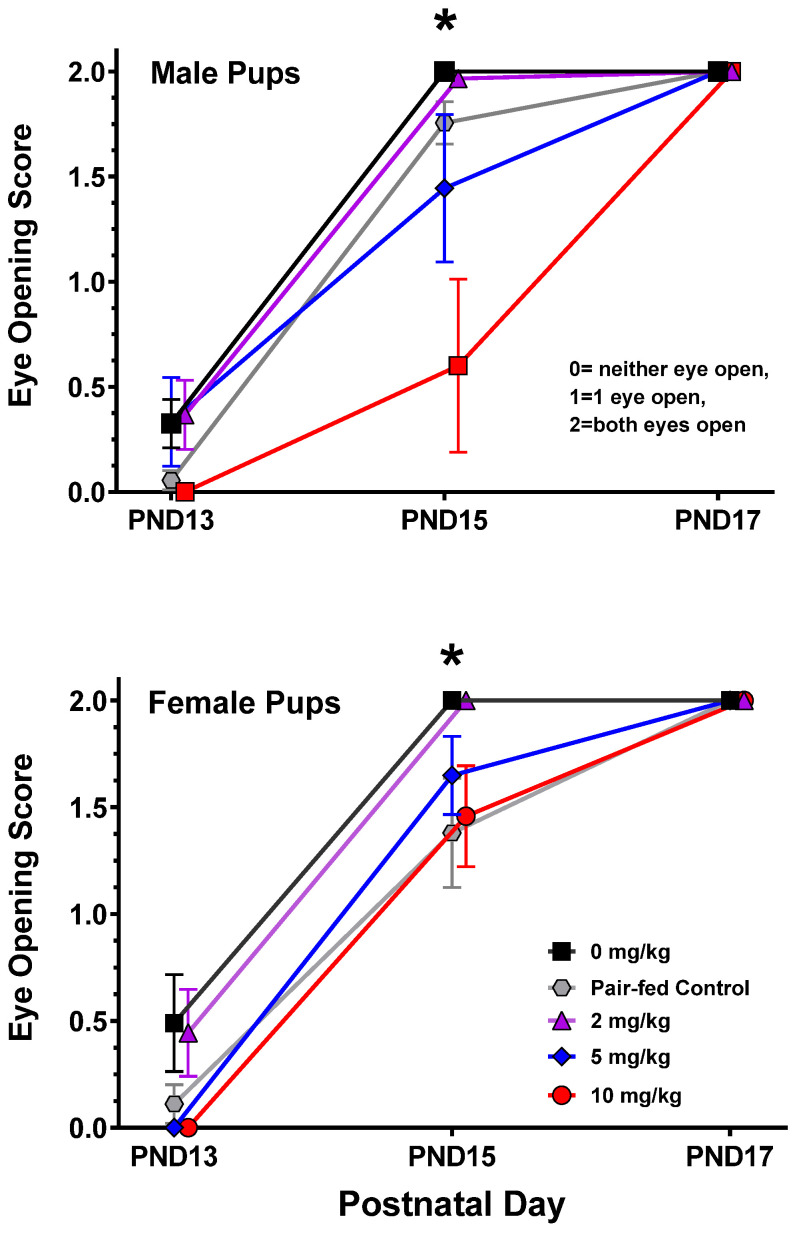
Mean eye opening score (±SEM) during the postnatal day 13–17 period. The 10 mg/kg males (top) and females (bottom) were significantly delayed on PND15. Both eyes were open/unsealed in all pups by PND17. For males, * 10 mg/kg < all other groups on PND15. For females, * control and 2 mg/kg > all other groups on PND15. N = 9, 5, 7, 5, 5 L for the 0 mg/kg, pair-fed, 2 mg/kg, 5 mg/kg, and 10 mg/kg groups, respectively.

**Figure 6 toxics-12-00311-f006:**
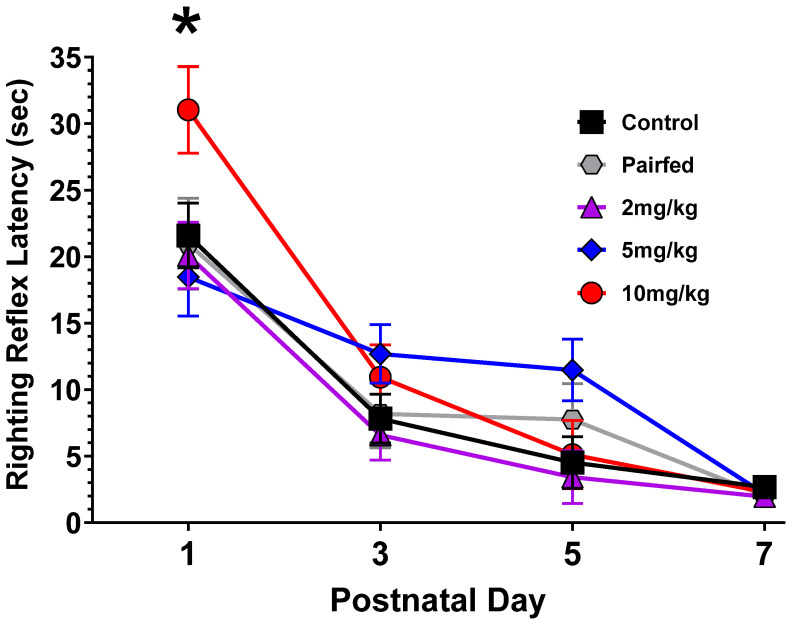
Mean latency (±SEM) in the pups re-establishing posture after being rolled on their side during the postnatal day 1–7 period. Data from male and female offspring have been averaged together. The latency for the 10 mg/kg group was significantly longer than the others on PND1. * 10 mg/kg > all other groups on PND1. N = 9, 5, 8, 6, 5 L for the 0 mg/kg, pair-fed, 2 mg/kg, 5 mg/kg, and 10 mg/kg groups, respectively.

**Figure 7 toxics-12-00311-f007:**
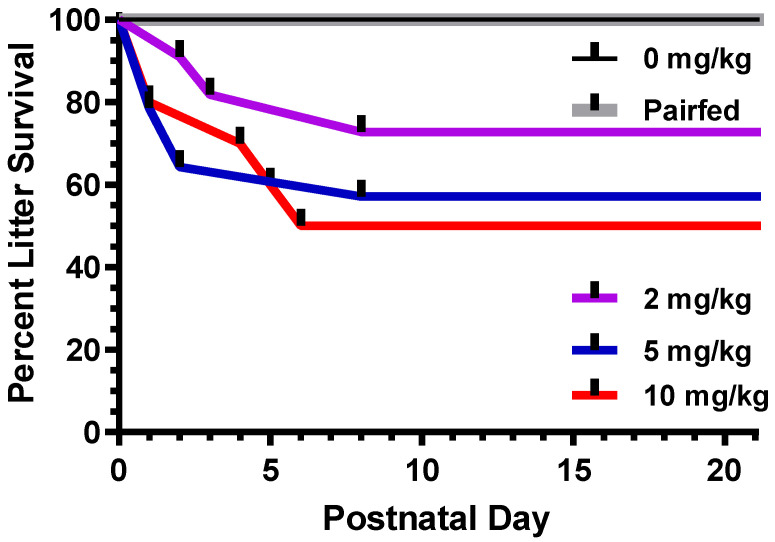
Attrition over the course of the preweaning period due to maternal neglect. The data indicate the loss of entire litters rather than individual pups. All litters in the control and pair-fed groups survived.

**Figure 8 toxics-12-00311-f008:**
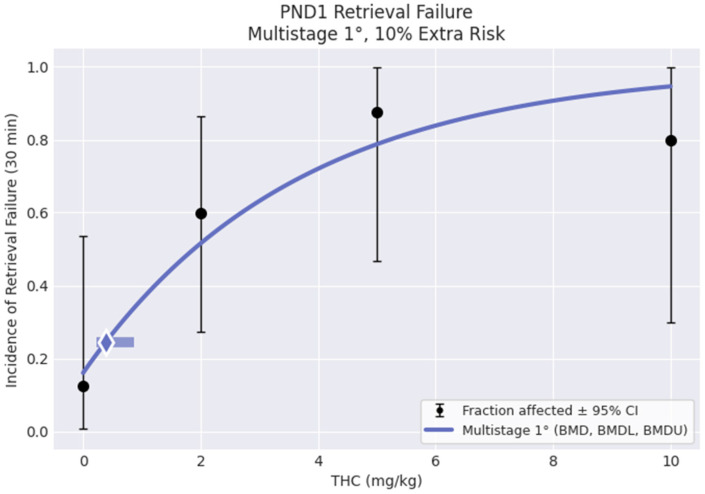
Output plot generated by USEPA BMDS online software, based on the incidence of complete litter retrieval failures in the control group (*n* = 8, incidence = 1), 2 mg/kg (*n* = 10, incidence = 6), 5 mg/kg (*n* = 8, incidence = 7), and 10 mg/kg (*n* = 5, incidence = 4) THC groups on postnatal day 1.

**Figure 9 toxics-12-00311-f009:**
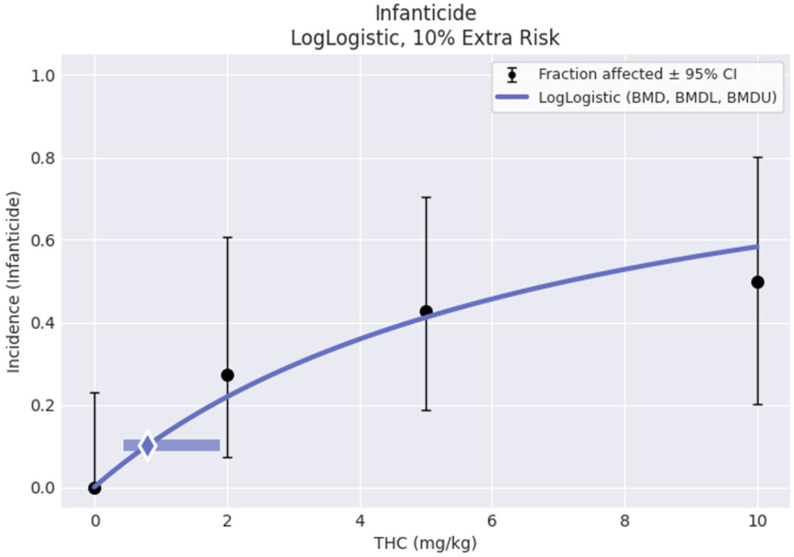
Output plot generated by USEPA BMDS online software, based on the incidence of complete litter attritions due to maternal neglect from PND1–21. Control group (*n* = 17, incidence = 0), 2 mg/kg (*n* = 11, incidence = 3), 5 mg/kg (*n* = 14, incidence = 6), and 10 mg/kg (*n* = 10, incidence = 5).

## Data Availability

The data presented in this study are available on request from the corresponding author.

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
