# Peer review of "Perinatal Tetrahydrocannabinol Compromises Maternal Care and Increases Litter Attrition in the Long–Evans Rat"

_toxics, 2024, doi:10.3390/toxics12050311_

Round 1
Reviewer 1 Report
Comments and Suggestions for Authors
The manuscript investigated the impact of perinatal THC exposure on maternal care and offspring development in rats. The study addresses a timely and relevant topic, given the increasing acceptance and use of cannabis. While the research design is sound and the results are intriguing, there are areas that require clarification and strengthening before publication.
1. I suggest adding a concluding statement about the significance and importance of this study in the final sentence of the abstract
2. I suggest shortening the introduction section as it is clear but a bit lengthy
3. I suggest avoiding a questioning tone in academic papers, especially in line 136.
4. I suggest that the last paragraph of the preface should briefly introduce the main content of the study.
5. I suggest that the authors provide a specific ethics statement number in their work. This would add credibility and transparency to their research.
6. The discussion of potential mechanisms underlying the observed effects could benefit from more depth.
7. The limitations of this study should be thoroughly discussed.
8. I suggest creating a schematic diagram to visually represent the main content of the study. This would help readers better understand the concept and processes involved in the research.
Reviewer 2 Report
Comments and Suggestions for Authors
Toxics-2959669_v1.
Peer review, England, 03rd of April 2024.
The research article by Carlson et alia examines the consequences of chronic all-trans-delta-9-THC administered to female rats from gestational day 1 to PND21. Authors used several groups: 0, 2, 5 and 10 mg/kg daily, as well as a pair-fed group serving as an additional control. The article reports significant effects of THC on dams’ body weight and latency to retrieve pups. Furthermore, THC impaired two developmental milestones: the righting reflex and time of eye opening. Authors also inferred from their results (pup mortality) THC doses that would induce a 10% risk of pup mortality at 0.8 and 0.4 mg/kg (respectively for benchmark dose and its lower bound).
The manuscript is well written and flows easily. It was very pleasant to read. Results and statistics are reported to high standards. There is no doubt that the results presented herein are of novelty and will be of interest to the field, for potential clinical and pre-clinical use. The study is well suited to the journal it is submitted in. Although there are several comments below, these only warrant minor revisions, as the below comments can be easily addressed. The manuscript will benefit from these small edits before publication can be recommended. However, please note that I am selecting “major revisions” in the reviewer form so that the rebuttal can be assessed.
Major comments:
1. Lines 12-13: the abstract needs to precisely mention delta-9 THC, not just THC. Please edit, as line 150 clearly states that delta-9 THC was used throughout this study.
2. Line 154: please refrain from using non-SI units. Why are authors writing “per 1 ml”? (/1ml). A widely used system of reporting concentrations is simply “X mg/ml”, not “X mg/1ml”. Please edit.
3. Section 2.2. Here, authors are expected to explicitly state the Ethical Approval number and the awarding body (generally, their local Ethics Committee). Please amend.
4. Section 2.2, line 165. Why have authors chosen to use a liquid-based diet for pregnant dams? Is this standard practice? Surely, there are perfectly adequate dry diet that have been formulated especially for gestation (such as A03 for example, see https://safe-lab.com/safe-wAssets/docs/product-data-sheets/diets/safe_a03_ds.pdf). Selection of a liquid-based diet needs to be justified. Furthermore, for animal welfare, have authors supplied the animals with materials for grinding and/or gnawing their teeth to prevent teeth overgrowth and malocclusion?
5. Section 2.2, lines 162-163. Please explicit cage dimensions and consider changing the term “shoebox”.
6. Section 2.5, line 210. Please specify what was the criteria for determine whether a litter was undersized or not.
7. Section 2.5, line 211. Please explicitly state the method of euthanasia.
8. Figure 2: please change the color scheme for all readers, including color-blind readers. In addition, it is now general practice to display individual points superimposed on bar charts. This comment applies to all subsequent bar chart figures.
9. Figure 4: please change the color scheme (as above). This comment applies to all subsequent figures.
10. Results section (general comment): how can authors differentiate whether the THC effects on pups are explained by the THC itself on pups (passed via the breastfeeding milk) or because of the effects of THC on the mothers (careless mothers)? Surely, both could have an effect? A small paragraph on this would be ideal for readers to put these results into perspective.
Minor comments:
1. Line 18: here, readers might be confused with the term “pup attrition”. Is this similar to pup mortality? If so, please change to the latter. Same comment for line 118. “Attrition” is fine elsewhere, but authors might want to be more specific in their abstract.
2. Line 35: not sure why authors have written “reproductive lifespan”. Do they mean reproductive period?
3. Lines 96-113. This paragraph is very interesting. Have oxytocin levels been measured in the dorsal raphe nucleus following THC exposure? If so, this could be a nice addition at the end of the section to show potential consequences of acute/chronic THC on mood and/or depression. This is just a suggestion.
4. Line 137: abnormal spacing. Also on line 355.
5. Line 179: a typo has been inserted in “by her 10mkg/kg”. If this is not a typo, then I am unsure of what this unit represents.
6. Line 215 (section 2.5): please check that the detection limit is 10 parts per billion (is this a typo?). If this is indeed ppb, please disregard this comment.
7. Line 222 (section 2.6), please use a lower p to define the p value threshold (p ≤ 0.05 instead of “P ≤ 0.05”), as reported throughout the results section.
8. Line 365 of the discussion. Does THC really accumulate? What are the pharmacokinetics of THC? Please see the following articles: PMID 3000569, PMID 31152723 and PMID 12648025. To prove bioaccumulation, THC has to possess specific pharmacokinetic properties. Please check. Maybe some elements to answer this can be found in Hutchings and colleagues (reference number 57, cited on lines 417-419). Did they see bioaccumulation?
Reviewer 3 Report
Comments and Suggestions for Authors
In the present manuscript the authors assessed the effects of perinatal exposure with the psychotropic phytocannabinoid THC in the maternal behavior and offspring development of rats. They found that the perinatal THC exposure affected maternal behavior as described by the enhanced latency to retrieve litter, which was not due to altered locomotor activity induced by perinatal THC exposure. THC (10 mg/kg) affected pup development as described by the delayed eye-opening score and by enhanced latency of righting reflex. Interestingly, litter survival was also affected by perinatal THC exposure. The authors conclude that perinatal THC exposure may affect neurodevelopment which is evident already at early postnatal days. Overall, the manuscript further confirms the potential long-lasting detrimental effects of perinatal THC exposure on neurodevelopment in rats (see Di Bartolomeo et al., 2023 doi: 10.3390/ijms24043907; Castelli et al., 2023 doi: 10.3390/pharmaceutics15020692; Di Bartolomeo et al., 2021 Pharmacol Res doi: 10.1016/j.phrs.2020.105357, Drazanova et al., 2019 doi: 10.1038/s41598-019-42532-z, which should be cited), providing some interesting and novel findings in terms of maternal behavior. However, I have several concerns with the current study.
A) Abstract. Methods, I suggest to add more details about the experimental design such day of start and duration of the treatment, as well as it should be better specify the p value of statistical analysis.
B) Introduction. I strongly suggest to add the following references supporting the detrimental long-lasting effect of perinatal THC exposure, which is evident at neonatal age: Di Bartolomeo et al., 2021 Pharmacol. Res. 164:105357. DOI: 10.1016/j.phrs.2020.105357, at adolescent age: Castelli et al., 2023 doi: 10.3390/pharmaceutics15020692 and at adulthood: Di Bartolomeo et al., 2023 doi: 10.3390/ijms240439072; Drazanova et al., 2019 doi: 10.1038/s41598-019-42532-z).
C) Materials and Methods. The number of animals per group should be described.
D) Discussion. The limitations of conducting their research are not shared in the discussion. The authors should present their study limitations, clarifying these points to the scientific research community. For instance, the authors did not perform molecular analysis as well as the phenotype of the rats was assessed just at neonatal age. Line 401 “the mechanisms responsible…” the authors cannot exclude that the delayed appearance of neonatal reflex may be due to the altered endocannabinoid signaling as previously described (see Di Bartolomeo et al., 2021 Pharmacol. Res. 164:105357. DOI: 10.1016/j.phrs.2020.105357)
E) Figure. Please add the symbol (i.e. *) based on the statistical significance
Conclusions: This paper does not deserve publication on Toxics as it stands. A complete revision of text is required before the paper can be admitted to a re-submission for publication.
Round 2
Reviewer 2 Report
Comments and Suggestions for Authors
The authors have addressed all my concerns.
I now recommend publication of the manuscript in its revised form.
Comments on the Quality of English Languageas above